# ADAM9 promotes type I interferon-mediated innate immunity during encephalomyocarditis virus infection

Lindsey E. Bazzone [1,2], Junji Zhu[3], Michael King[1], GuanQun Liu [3], Zhiru Guo [1], Christopher R. MacKay[1], Pyae P. Kyawe[1], Natasha Qaisar[1], Joselyn Rojas-Quintero[4], Caroline A. Owen [4], Abraham L. Brass[5], William McDougall[5], Christina E. Baer [5], Timothy Cashman[6], Chinmay M. Trivedi [6], Michaela U. Gack [3], Robert W. Finberg[1,7,8] & Evelyn A. Kurt-Jones [1,7] ✉

Viral myocarditis, an inflammatory disease of the heart, causes significant morbidity and mortality. Type I interferon (IFN)-mediated antiviral responses protect against myocarditis, but the mechanisms are poorly understood. We previously identified A Disintegrin And Metalloproteinase domain 9 (ADAM9) as an important factor in viral pathogenesis. ADAM9 is implicated in a range of human diseases, including inflammatory diseases; however, its role in viral infection is unknown. Here, we demonstrate that mice lacking ADAM9 are more susceptible to encephalomyocarditis virus (EMCV)-induced death and fail to mount a characteristic type I IFN response. This defect in type I IFN induction is specific to positive-sense, single-stranded RNA (+ ssRNA) viruses and involves melanoma differentiation-associated protein 5 (MDA5)−a key receptor for +ssRNA viruses. Mechanistically, ADAM9 binds to MDA5 and promotes its oligomerization and thereby downstream mitochondrial antiviral-signaling protein (MAVS) activation in response to EMCV RNA stimulation. Our findings identify a role for ADAM9 in the innate antiviral response, specifically MDA5-mediated IFN production, which protects against virus-induced cardiac damage, and provide a potential therapeutic target for treatment of viral myocarditis.

Encephalomyocarditis virus (EMCV) is a picornavirus and a member of the cardiovirus family. EMCV infects multiple organs, including the heart, brain, and pancreas. In pigs, rodents, and non-human primates, EMCV infection can cause severe myocarditis as well as encephalitis, diabetes, and reproductive pathology[1]. EMCV infection of mice is a well-established model for viral myocarditis, congestive heart failure, and dilated cardiomyopathy[2]. EMCV is an important veterinary pathogen that has caused large zoonotic epidemics that have

[1]Department of Medicine, Division of Infectious Diseases and Immunology, University of Massachusetts Chan Medical School, Worcester, MA, USA. [2]Department of Medicine, Division of Pulmonary and Critical Care Medicine, University of Massachusetts Chan Medical School, Worcester, MA, USA. [3]Florida Research and Innovation Center, Cleveland Clinic, Port St Lucie, FL, USA. [4]Division of Pulmonary and Critical Care Medicine, Brigham and Women's Hospital and Harvard Medical School, Boston, MA, USA. [5]Department of Microbiology and Physiological Systems, University of Massachusetts Chan Medical School, Worcester, MA, USA. [6]Department of Medicine, Division of Cardiovascular Medicine, University of Massachusetts Chan Medical School, Worcester, MA, USA. [7]Program in Innate Immunity, University of Massachusetts Chan Medical School, Worcester, MA, USA. [8]Deceased: Robert W. Finberg. ✉e-mail: Evelyn.Kurt-Jones@umassmed.edu

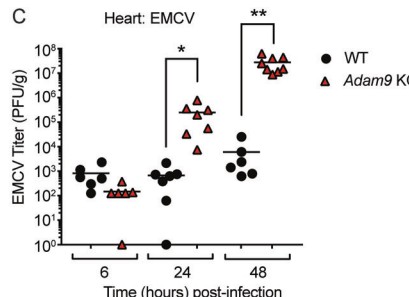

**Fig. 1 | *Adam9* KO mice are highly susceptible to EMCV infection of the heart.**
**A** WT (*n* = 9) and *Adam9* KO (*n* = 6) mice were infected by intraperitoneal (i.p.) injection of $10^5$ PFU of EMCV and survival was assessed. *Adam9* KO mice demonstrated accelerated mortality compared to WT mice in response to EMCV infection (***$P$ = 0.0002, Mantel−Cox survival analysis). Survival data represents two independent experiments. WT and *Adam9* KO mice were infected i.p. with $10^5$ PFU of EMCV and EMCV titers were measured by plaque assay at 6, 24, and 48 h p.i. in serum (**B**) and hearts (**C**). **B** EMCV titers in the serum were delayed in *Adam9* KO

compared to WT mice at 6 h pi. (**$P$ = 0.0064) but were higher in *Adam9* KO compared to WT mice by 48 h p.i. (**$P$ = 0.0027) by a two-tailed, unpaired $t$-test. **C** In addition, EMCV titers in the heart were also higher in *Adam9* KO compared to WT mice at 24 h p.i. (*$P$ = 0.0318) and 48 h p.i. (**$P$ = 0.0046) by a two-tailed, unpaired $t$-test. **B**, **C** All data are representative of three independent experiments with similar results. Each symbol represents an individual mouse (*n* = 6 for 6 h p.i.; *n* = 7 for 24 h p.i.; *n* = 6 for WT and *n* = 8 for KO for 48 h p.i.), and small horizontal lines indicate the mean. Source data are provided in the Source Data file.

decimated domestic animal populations, particularly pigs[1]. Symptomatic infections in humans are rare and usually mild; however, human cells are very susceptible to EMCV infection and are rapidly killed by the virus. Human infections are associated with occupational exposure to animals, suggesting that animal-to-human transmission of EMCV occurs, making the zoonotic potential of EMCV transmission to humans a major concern.

Type I interferons (IFNs), including IFN-β, are produced early in the immune response to viral infection, inducing antiviral effects in target cells and subsequently mediating a variety of downstream innate and adaptive immune responses. Type I IFNs are critical for early control of EMCV infection as well as infection with other picornaviruses, such as coxsackievirus. Induction of the early IFN response depends on rapid virus recognition by the innate immune system and is essential to controlling infection and spread. RIG-I-like receptors (RLRs) have emerged as an important class of cytosolic sensor proteins that initiate type I IFN (and other cytokine) responses following virus infection[3]. Within the RLR family, MDA5 (melanoma differentiation-associated protein 5) senses cytosolic RNA of members of the *Picornaviridae* family, including EMCV and coxsackievirus B3 (CVB3)[4–6].

To understand the complex biology of EMCV infection and pathogenesis, we employed in our previous studies a functional genomics approach to identify genes responsible for EMCV-induced lytic infection of human cells. Using a genome-wide CRISPR-Cas9 screen, we identified A Disintegrin And Metalloproteinase domain 9 (ADAM9) as a major EMCV-dependency factor[7]. ADAMs are a family of transmembrane metalloproteinases that play important roles in growth factor and cytokine signaling, as well as cell-to-cell signaling, adhesion, and extracellular matrix remodeling[8–17]. ADAM9 consists of several highly conserved domains: a pro-domain that is removed during protein maturation, a metalloproteinase domain containing the enzyme active site, a disintegrin domain that can potentially mediate interactions with integrins on the cell surface, a Cys-rich domain of unknown function, an epidermal growth factor (EGF)-like domain of unknown function, a transmembrane domain, and a cytoplasmic tail containing several Src homology 3 (SH3) motifs[18–23]. Approximately half of the ADAM family members, including ADAM9, have proteolytic capabilities that modulate the activity of cytokines, chemokines, and growth factors, as well as their associated receptors and cell adhesion molecules[9,17,18,24]. ADAMs have been implicated in a range of human diseases, including certain types of cancer, inflammatory diseases, wound healing, and microbial infections; however, very little is known about the role of ADAMs in viral infection.

In the present study, we investigated the role of ADAM9 in picornavirus infection using *Adam9* KO mice and an infection model of EMCV-induced myocarditis. We discovered that ADAM9 deficiency enhanced cardiac infection with EMCV, resulting in rapid death. Our results show that in the absence of ADAM9, EMCV infection fails to trigger a systemic antiviral response, which ultimately leads to uncontrolled viral replication in the heart resulting in fatal cardiac damage. In contrast, wild-type (WT) mice encoding intact *Adam9* mounted an early, robust IFN-β response that is cardioprotective, limiting viral replication and direct cardiac damage. Furthermore, we demonstrate that ADAM9 regulates innate immunity mediated by MDA5, which is a cytosolic sensor that plays an important role in recognizing the RNA of EMCV and other picornaviruses. Specifically, ADAM9 interacts with MDA5 facilitating its oligomerization and subsequent MAVS activation, thereby promoting the antiviral IFN response in a manner independent of ADAM9's catalytic activity. Our results establish a mechanism of ADAM9-dependent protection of the heart during viral infection, unveiling a previously undiscovered role for ADAM9 in the type I IFN response to cytosolic viral RNA. To our knowledge, this study is the first to identify ADAM9 as a key regulator of the innate immune response in virus-induced myocarditis.

## Results

### ADAM9 deficiency increases mortality and cardiac viral load following EMCV infection

Using a CRISPR-Cas9 screen, we previously identified ADAM9 as a major susceptibility factor for EMCV infection of both animal and human cells[7]. To investigate the role of ADAM9 in picornavirus infection in vivo, we used *Adam9* KO mice and an infection model of EMCV-induced myocarditis. To determine the impact of ADAM9 deficiency on survival during EMCV infection, we assessed mortality in EMCV-infected WT and *Adam9* KO mice. EMCV-infected WT mice died at 72−96 h post-infection (p.i.). In contrast, *Adam9* KO mice infected with EMCV died earlier than WT mice, with sudden death at 48−60 h p.i. (Fig. 1A). Analysis of EMCV loads in the heart revealed that *Adam9* KO mice had significantly higher serum (Fig. 1B) and cardiac (Fig. 1C) titers of EMCV compared to WT mice as early as 24 h p.i. The increased viral load in the hearts of EMCV-infected *Adam9* KO mice led us to investigate whether the earlier mortality seen in these mice was due to infection-mediated cardiac pathology.

### ADAM9 is cardioprotective during in vivo EMCV infection

The high EMCV titers in cardiac tissues from *Adam9* KO mice compared to WT mice suggested that EMCV replication is enhanced in the absence of ADAM9. To determine the impact of ADAM9 deficiency on

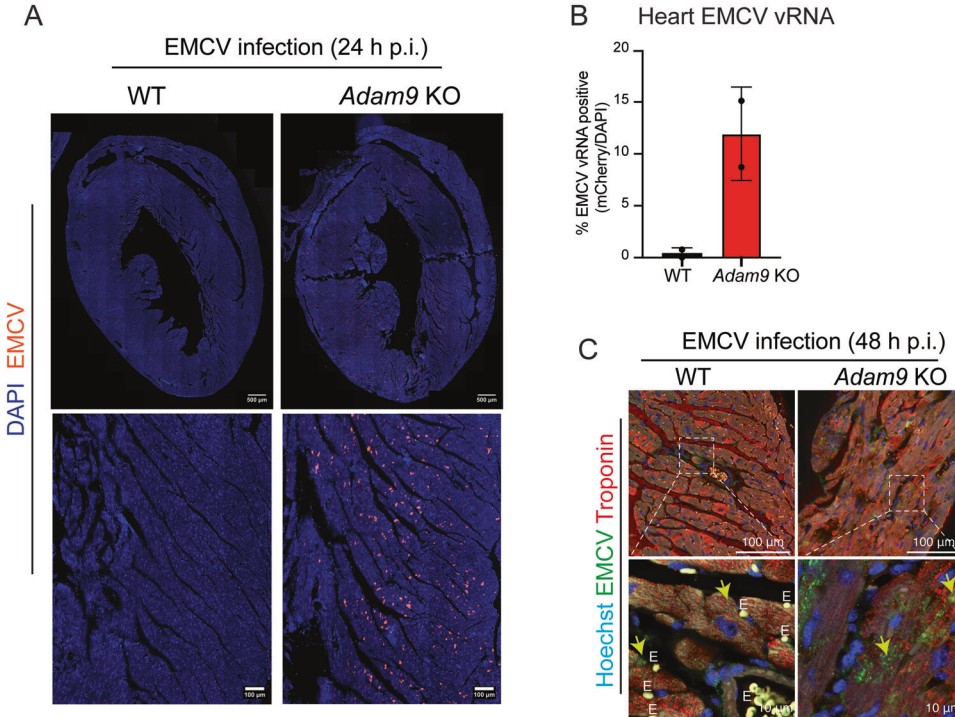

**Fig. 2 | ADAM9 deficiency enhances viral infection of cardiac tissue. A** EMCV genomes in the hearts from WT and *Adam9* KO mice visualized by RNAscope™. Representative images of hearts from WT (*n* = 2) or *Adam9* KO (*n* = 2) mice 24 h after i.p. injection with $10^5$ PFU of EMCV. Samples were fixed with 4% PFA, paraffin-embedded, and cut into 8-micron sections on regular slides. EMCV genomes in the tissues were detected by RNAscope™ (following the Multiplex Fluorescent Reagent Kit v2 Assay from ACD) using an EMCV-specific RNAscope™ probe-C2 mCherry that detects the EMCV genomes. Nuclei were stained with DAPI and visualized with fluorescence microscopy. Pixels = 325 nm × 325 nm. **B** Quantification of mCherry positive (EMCV vRNA) per DAPI positive nuclei in whole heart tissue scan from (**A**).

Higher levels of EMCV genomes were detected throughout all regions of the heart in *Adam9* KO hearts (11.92% mCherry positive foci per DAPI) compared to WT hearts (0.40% mCherry positive foci per DAPI). Bar graphs represent the mean ±SD of mCherry positive DAPI+ nuclei. Number of DAPI+ nuclei analyzed per condition: WT + EMCV, *n* = 156532; *Adam9* KO + EMCV, *n* = 118895. **C** Immunohistochemical detection of EMCV (green) and troponin (red) in WT and *Adam9* KO hearts 48 h p.i. with $10^5$ PFU of EMCV. EMCV-infected cardiomyocytes (arrows). Nuclei stained with Hoechst (blue). Erythrocytes marked "E." Source data are provided in the Source Data file.

viral replication in the heart, we visualized EMCV genomes in cardiac tissue sections using RNAscope™ and quantitated EMCV genomes in whole heart-mounted paraffin-embedded tissue sections. The hearts of WT mice showed a modest, focal infection of cardiac tissue. In contrast, cardiac tissue from *Adam9* KO mice showed diffuse, widespread infection with high levels of EMCV RNA genomes detected throughout all regions of the heart (Fig. 2A, B). Immunohistochemical analysis of cardiac tissue also demonstrated increased EMCV infection in *Adam9* KO compared to WT cardiomyocytes (stained with cardiac troponin I, cTNI, a marker of cardiac damage in humans and mice) (Fig. 2C). Thus, ADAM9 deficiency resulted in fulminant EMCV infection of the myocardium. Interestingly, inflammatory cell infiltrates were not observed in either WT or *Adam9* KO cardiac tissues following EMCV infection. Severe histopathologic changes in the absence of inflammatory cell infiltrate were consistently seen in *Adam9* KO mice but not in WT mice, suggesting cardiac damage/dysfunction due to direct viral damage as the primary cause of early EMCV-induced mortality observed in *Adam9* KO mice.

To determine the mechanism(s) underlying the earlier mortality of *Adam9* KO mice after EMCV infection, we assessed cardiac histopathology in EMCV-infected WT and *Adam9* KO mice. Post-mortem cardiac histopathology demonstrated that *Adam9* KO mice developed a severe dilated cardiomyopathy with increased interstitial edema between the myofibers, interrupting the tissue structure (Fig. 3A, B). Serum levels of cardiac troponin I (cTNI), a marker of cardiomyocyte necrosis, were significantly elevated in EMCV-infected *Adam9* KO mice compared to WT mice, indicating that EMCV-induced cardiac damage was amplified in the absence of ADAM9 (Fig. 3C).

To further investigate cardiac disease in EMCV-infected WT and *Adam9* KO mice, we performed immunohistochemistry analyzing troponin-T (to identify cardiomyocytes) and wheat germ agglutinin (WGA) (to delineate cell membranes and thereby cell damage). We noted damage to cardiomyocytes, with defective sarcomeric structures and release of troponin into the extracellular space in EMCV-infected *Adam9* KO hearts at 48 h p.i. (Fig. 3D).

Taken together, *Adam9* KO mice had increased viral replication in the heart resulting in acute myocardial damage, suggesting that ADAM9 may play a protective role during viral infection of the heart.

### ADAM9 mediates innate immune responses to picornavirus infection

*Adam9* KO mice demonstrated unexpected sensitivity to EMCV infection and were also prone to developing fulminant myocarditis resulting in early mortality as compared to infected WT controls. Our data indicate that ADAM9 expression protects the heart from overwhelming EMCV infection, suggesting that ADAM9 may play a role in promoting innate immunity to EMCV infection. To understand the mechanism(s) of cardiac disease and early mortality in EMCV-infected *Adam9* KO mice, we tested the hypothesis that ADAM9 may be involved in the innate antiviral response. To this end, we measured IFN-β and other cytokine levels in uninfected and EMCV-infected WT and *Adam9* KO mice. WT mice expressed IFN-β in heart tissues in response to EMCV infection. Furthermore, IFN-β was also detected in the serum of EMCV-infected WT mice; in stark contrast, serum levels of IFN-β were undetectable in *Adam9* KO mice (Fig. 4A). IFN-β production was also substantially reduced in the heart tissues of EMCV-infected *Adam9*

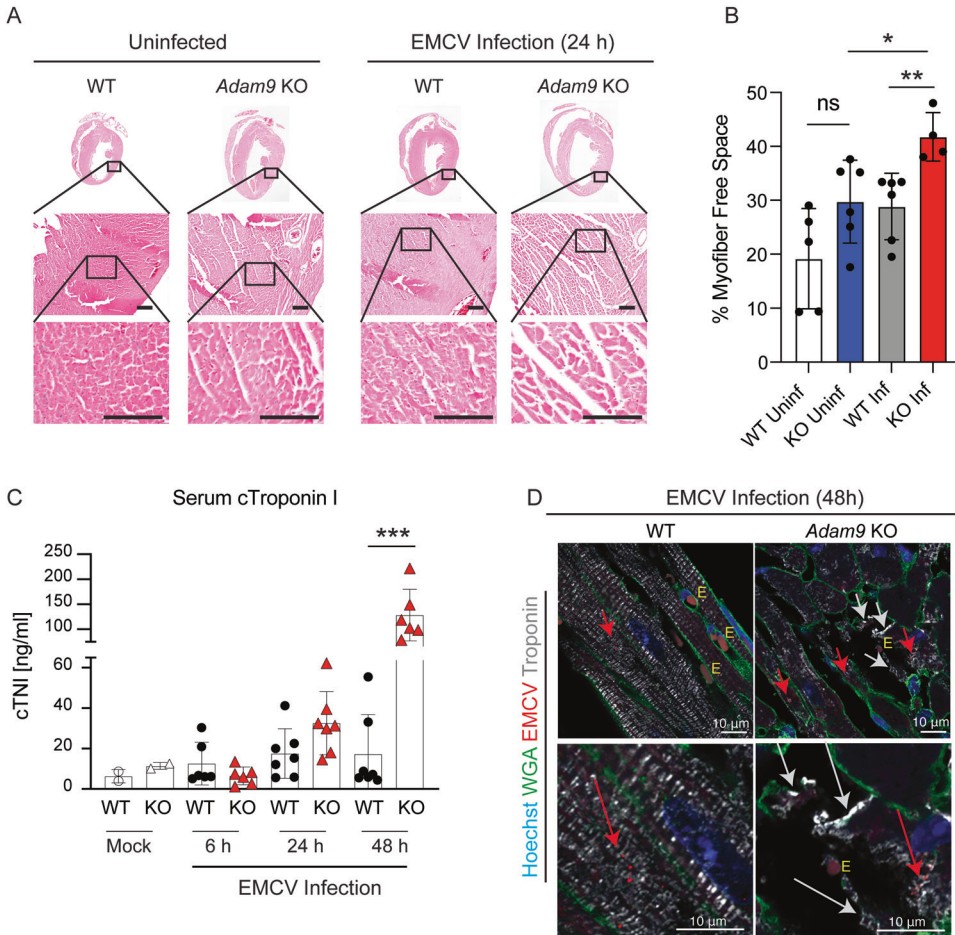

**Fig. 3 | EMCV infection of *Adam9* KO mice induces severe myocardial damage.**
**A** Representative images of hematoxylin and eosin (H&E)-stained heart sections from WT and *Adam9* KO mice uninfected or infected i.p. with 10⁵ PFU of EMCV. Scale bar = 100 µm. **B** Infected *Adam9* KO mice (*n* = 4) exhibit a significant increase in extracellular space between myofibers compared to uninfected *Adam9* KO mice (*n* = 6) (**P* = 0.0238) and infected WT mice (*n* = 6) (***P* = 0.0074), two-tailed, unpaired *t*-test. Error bars represent mean ± SD. Extracellular space was quantified by creating a binary image from each H&E image (*n* = 4–6 fields per genotype) at 20× magnification and manually outlining each myofiber. The percentage of white pixels (extracellular space) was calculated from an inverted binary image within each myofiber region of interest and averaged to calculate an average percentage of myofiber-free space per condition (6–12 myofibers per condition). Note: Uninfected *Adam9* KO mice and EMCV-infected WT mice each exhibit a trend towards increased intramyofiber spacing compared to uninfected WT mice (*n* = 5), but

(unlike the infected *Adam9* KO mice) these changes are statistically not significant (ns) (*P* = 0.0686; two-tailed, unpaired *t*-test). **C** WT and *Adam9* KO mice were mock-infected or infected i.p. with 10⁵ PFU of EMCV per mouse, and serum levels of cardiac troponin I (cTNI) were measured by ELISA at 6, 24, and 48 h p.i. Each symbol represents an individual mouse (mock-infected *n* = 2 mice each for WT and *Adam9* KO mice; infected WT and *Adam9* KO mice: *n* = 6 for 6 h p.i. and *n* = 7 for 24 h p.i. for both WT and *Adam9* KO mice; *n* = 7 WT mice and *n* = 6 *Adam9* KO mice for 48 h p.i.). Bar graphs indicate the mean ±SD, ****P* = 0.0002 (two-tailed, unpaired *t*-test). **D** Representative confocal immunofluorescence imaging of heart tissue from WT mice and *Adam9* KO mice 48 h p.i. with 10⁵ PFU of EMCV (red) showing EMCV-infected cardiomyocytes (red arrows) and defective sarcomeric structures (white arrows) with troponin (white) leakage from damaged cardiomyocytes in *Adam9* KO but not WT hearts. Wheat germ agglutinin (WGA, green), Hoechst (cell nuclei, blue). Erythrocytes marked "E." Source data are provided in the Source Data file.

KO mice compared to WT mice (Fig. 4B). Collectively, these results indicated that *Adam9* KO mice failed to produce IFN-β despite the presence of higher viral loads in the circulation and within cardiac tissues compared to WT mice (see Fig. 1).

To determine whether the ADAM9-dependent IFN-β response is unique to EMCV, or if ADAM9 is involved in the IFN response to picornaviruses in general, we infected WT and *Adam9* KO mice with coxsackievirus B3 (CVB3). Similar to EMCV, CVB3 infection resulted in a significantly blunted IFN-β response in *Adam9* KO mice compared to WT control mice (Fig. 4C). Thus, ADAM9 mediates the type I IFN response to EMCV and other picornaviruses.

We also assessed the inflammatory cytokine response to EMCV infection. Similar to IFN-β, macrophage inflammatory protein-1α (MIP-1α) and MIP-1β were rapidly induced in WT mice after EMCV infection, and peak levels were detected in serum at 6 h p.i. (Fig. 5A, B). Furthermore, in the serum of WT mice, IL-6 and RANTES levels peaked at 24 h p.i. (Fig. 5C, D), while MCP-1 and CXCL1 levels continued to

increase up to 48 h p.i. (Fig. 5E, F). The levels of MIP-1α, MIP-1β, IL-6, RANTES, MCP-1, and CXCL1 were significantly reduced in *Adam9* KO compared to WT mice after infection (Fig. 5). These results suggest that the early and late IFN-β and pro-inflammatory cytokine responses to EMCV infection (and possibly other picornaviruses as demonstrated with CVB3) depend on ADAM9 because such responses are either absent or delayed in *Adam9* KO mice compared to WT controls. Furthermore, we surmised that the deficient innate immune response to viral infection in *Adam9* KO mice resulted in an increased viral load and accelerated mortality seen in these animals.

### ADAM9 is critical for the IFN-β response to intracellular viral RNA

Our in vivo studies clearly demonstrated that EMCV replicated to high levels in *Adam9*-deficient cardiac cells; thus, ADAM9 is not the cardiac entry receptor for EMCV; rather ADAM9 is necessary for the IFN-β response of cardiac cells to EMCV. To further investigate the

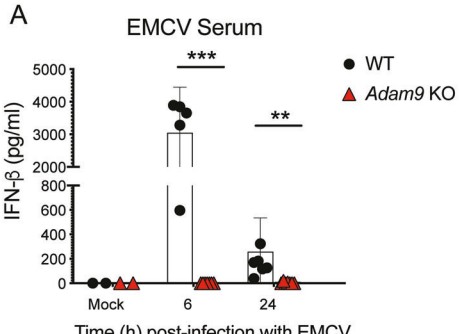

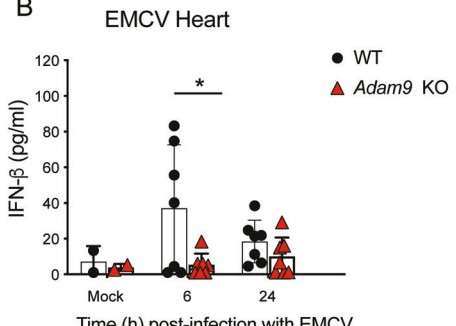

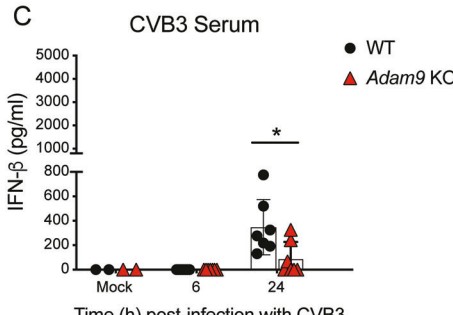

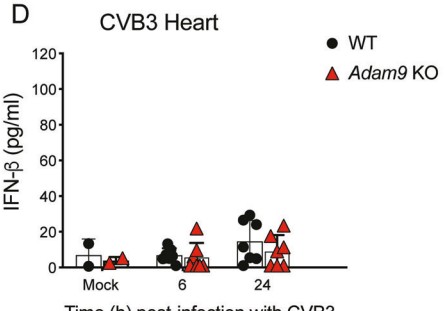

**Fig. 4 | ADAM9 is required for the IFN-β response to EMCV and Coxsackievirus infection in vivo. A–D** WT and *Adam9* KO mice were mock-infected or infected i.p. with $10^5$ PFU of EMCV or $10^5$ PFU of coxsackievirus B3 (CVB3). Serum and hearts from infected mice were collected at 6 and 24 h p.i., and IFN-β production was measured by ELISA. IFN-β levels in **A** serum of mock- ($n = 2$) versus EMCV-infected WT ($n = 5$ for 6 h p.i. and $n = 6$ for 24 h p.i.) and *Adam9* KO ($n = 7$ for 6 h p.i. and $n = 7$ for 24 h p.i.) mice and **B** hearts of mock- ($n = 2$) versus EMCV-infected ($n = 7$ for 6 h p.i. and 24 h p.i.) WT and *Adam9* KO mice. IFN-β levels in **C** serum and **D** hearts of mock- ($n = 2$) versus CVB3-infected ($n = 7$ at 6 h p.i. and 24 h p.i.) WT and *Adam9* KO mice. Each symbol represents an individual mouse; vertical bars indicate the mean ± SD. **A** \*\*\**P* = 0.0001, \*\**P* = 0.0011; **B** \**P* = 0.0390; **C** \**P* = 0.0241 (two-tailed, unpaired *t*-test). All data are representative of at least three independent experiments with similar results. Source data are provided in the Source Data file.

mechanism of ADAM9-mediated control of IFN-β responses, we examined the impact of ADAM9 on the IFN-β response of primary cells in vitro. Primary lung fibroblasts (LFs) express *IFNB1* mRNA and secrete IFN-β protein when infected with EMCV. In contrast to cardiac cells, ADAM9 acts as an entry receptor for EMCV on fibroblasts[7]. Therefore, these cells are not suitable for analyzing antiviral innate responses to EMCV because the failure of *Adam9* KO LFs to produce IFN-β following authentic EMCV infection likely reflects the failure of live EMCV to enter and replicate in these cells.

As a positive-strand RNA, the viral RNA (vRNA) genome of EMCV can replicate when introduced into the cell cytosol by transfection yielding high titers of infectious viral progeny. Importantly, when EMCV vRNA was transfected into cells, we noted that the requirement of ADAM9 for EMCV entry into fibroblasts was bypassed[7], and both WT and *Adam9* KO LFs produced infectious viral progeny (Fig. 6A). When fibroblast cells were transfected with low amounts of vRNA, we observed a reduced yield of viral progeny from *Adam9* KO compared to WT, likely reflecting the spread of infection to bystander fibroblasts in cultures of WT, but not *Adam9* KO, cells. Nevertheless, EMCV replicated to high levels in vRNA-transfected *Adam9* KO cells. Thus, as we previously noted[7], ADAM9 is not necessary for virus replication inside cells.

Transfecting EMCV vRNA into the cytosol of cells activates the production of IFN-β via MDA5[6,25,26], while transfection with rabies virus leader (RABV-Le) RNA or infection with Sendai virus (SeV) or vesicular stomatitis virus (VSV) activates IFN-β via RIG-I[3,27,28]. Therefore, we next assessed whether ADAM9 is necessary for the IFN-β response to cytosolic RNA. LFs from WT and *Adam9* KO mice were transfected with EMCV vRNA. Remarkably, EMCV vRNA-

induced IFN-β responses in WT but not *Adam9* KO LFs (Fig. 6B), despite the recovery of infectious viral progeny from both WT and *Adam9* KO cells (Fig. 6A). Additionally, *Adam9* KO LFs also failed to produce the pro-inflammatory cytokine IL-6 in response to EMCV vRNA (Fig. 6C).

The defect in IFN-β induction in *Adam9* KO cells was specific to EMCV vRNA as SeV elicited robust IFN-β induction, which was similar to responses in WT cells (Fig. 6D). Furthermore, the IFN response in cells stimulated with RIG-I ligands, i.e. RABV-Le RNA (Fig. 6E) and VSV vRNA (Fig. 6F) was also not affected by ADAM9 deficiency. Importantly, rescue of ADAM9 in KO cells restored the IFN-β response to EMCV vRNA (Fig. 6G). The failure of *Adam9* KO cells to respond to cytosolic EMCV vRNA revealed a critical role for ADAM9 in IFN-β-mediated host defense independent of its role as an entry receptor or host dependency factor.

### ADAM9 mediates type I IFN induction via MDA5

Previously, we and others established that the type I IFN response to EMCV infection and EMCV vRNA challenge is driven by activation of the cytosolic sensor MDA5 and not RIG-I[6,27,29]. MDA5 is also critical for the induction of the type I IFN response to CVB3 infection[5]. Our data demonstrated that ADAM9 is essential for IFN-β induction during EMCV (Fig. 4A, B) and CVB3 (Fig. 4C) infection in vivo and to EMCV vRNA challenge in vitro (Fig. 6B), but ADAM9 was not required for the IFN response to RIG-I ligands (Fig. 6D, E, F). Thus, we hypothesized that ADAM9 is specifically involved in mediating the MDA5-dependent innate antiviral response.

To determine a relationship between ADAM9 and MDA5, primary normal human lung fibroblasts (NHLFs) transfected with

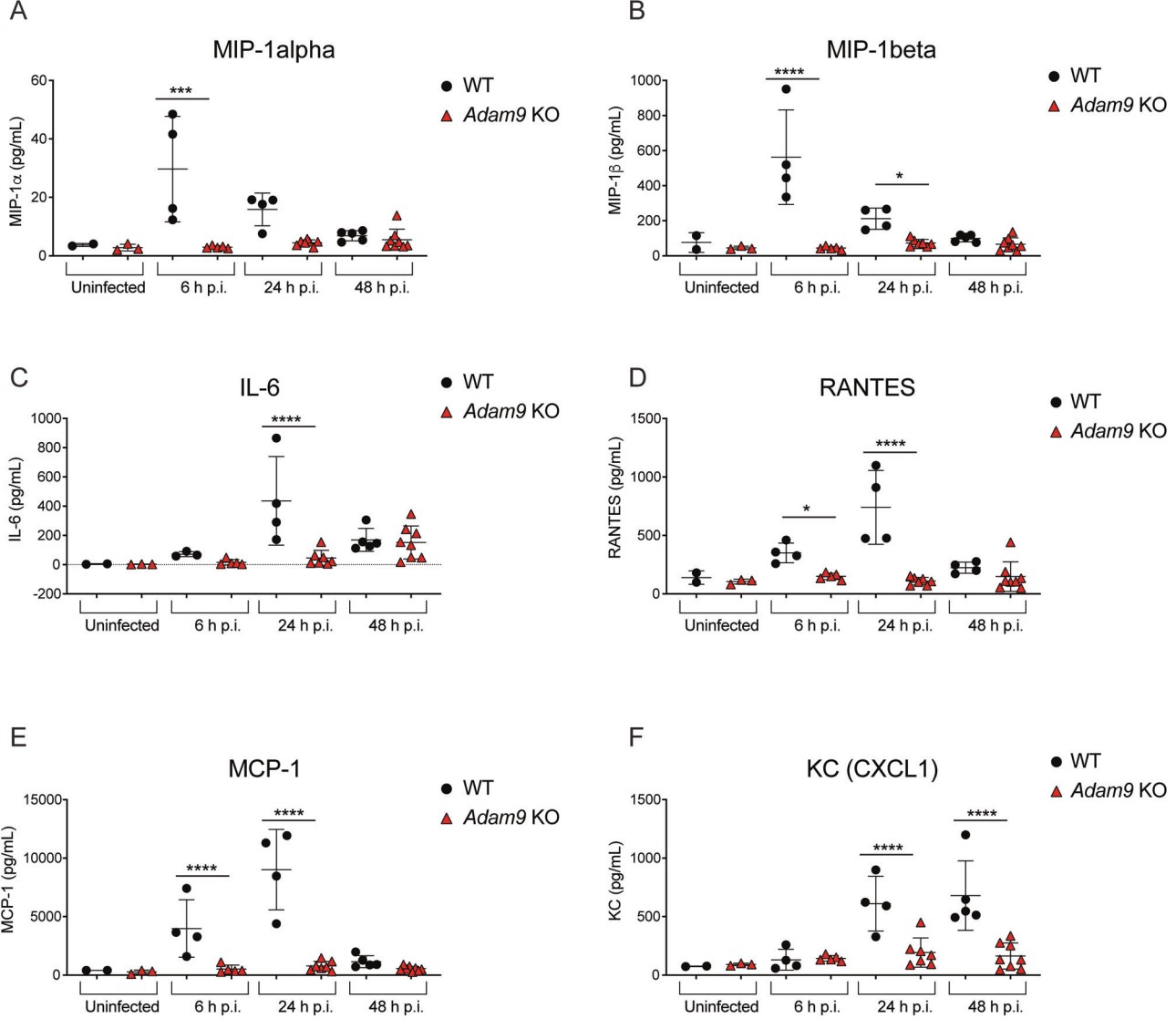

**Fig. 5 | ADAM9 is essential for the innate immune response to EMCV infection in vivo.** WT mice ($n = 2$ for uninfected; $n = 4$ for 6 h p.i., except $n = 3$ for IL-6; $n = 4$ for 24 h p.i.; $n = 5$ for 48 h p.i., except $n = 4$ for RANTES) and *Adam9* KO mice ($n = 3$ for uninfected; $n = 5$ for 6 h p.i.; $n = 7$ for 24 h p.i.; $n = 8$ for 48 h p.i.) were infected i.p. with $10^5$ PFU of EMCV, and serum was collected at 6, 24, and 48 h p.i. Serum cytokine levels were measured by Luminex multiplex analysis with each sample run in duplicate. Each symbol represents an individual mouse; small horizontal lines indicate the mean ± SD. Two-way ANOVA with Tukey's multiple comparisons test was used to determine statistical significance; **A** ***$P = 0.0002$; **B** *$P = 0.0418$, ****$P < 0.0001$; **C** ****$P < 0.0001$; **D** *$P = 0.0101$, ****$P < 0.0001$; **E** ****$P < 0.0001$; **F** ****$P < 0.0001$. Source data are provided in the Source Data file.

either non-targeting control siRNA (si-C) or ADAM9-specific siRNA (si-ADAM9) were challenged with EMCV RNA, and IFN-β and IL-6 transcripts were assessed by qRT-PCR. EMCV RNA challenge of ADAM9-depleted cells significantly reduced IFN-β and IL-6 transcripts compared to control cells (Fig. 7A, B). To confirm that ADAM9 promotes the antiviral response via MDA5, we determined the effect of ADAM9 deficiency on signaling induced by ectopic expression of either FLAG-MDA5 or FLAG-RIG-I. As compared to WT control cells, the induction of *IFNB1* mRNA was markedly reduced in ADAM9 KO cells expressing FLAG-MDA5, whereas there was no effect on IFN-β induction mediated by FLAG-RIG-I (Fig. 7C). Taken together, these results show that ADAM9 promotes MDA5-, but not RIG-I-, mediated antiviral signal transduction.

## ADAM9 binds to MDA5 and promotes its oligomerization and downstream MAVS activation

We next sought to understand the molecular mechanism(s) by which ADAM9 facilitates MDA5-mediated IFN induction. We first examined

whether ADAM9 interacts with MDA5. Co-immunoprecipitation (co-IP) experiments showed that murine MDA5 (mMDA5) binds to both murine ADAM9 (mADAM9) WT and its catalytically inactive mutant (E→A) (Fig. 8A). In accord, ectopically expressed human MDA5 (hMDA5) also efficiently bound to mADAM9 WT and the E→A mutant (Fig. 8B), suggesting that ADAM9 binds to MDA5 in a manner independent of its catalytic activity. Furthermore, endogenous ADAM9 readily co-immunoprecipitated with endogenous MDA5 upon EMCV RNA stimulation, while no interaction was observed between the two proteins in uninfected conditions (Fig. 8C). Importantly, MDA5 interacted specifically with ADAM9, but not with ADAM10, ADAM12 or ADAM17 (Fig. 8D).

As vRNA-induced MDA5 oligomer formation is essential for IFN induction via activation of the RLR-adaptor protein MAVS[3], we next investigated the effect of ADAM9 on MDA5 oligomerization. Semi-Denaturing Detergent Agarose Gel Electrophoresis (SDD-AGE) analysis showed that ectopic expression of mADAM9 promoted MDA5 oligomerization upon EMCV RNA stimulation (Fig. 8E). In line

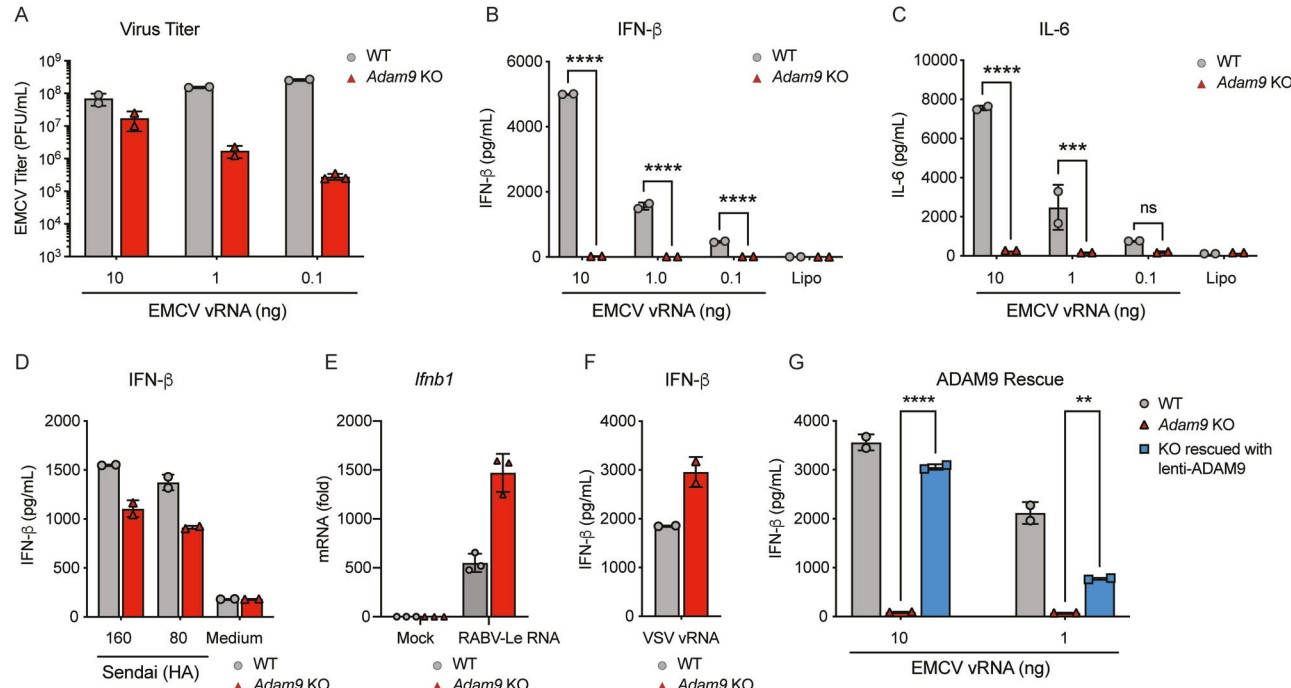

**Fig. 6 | ADAM9 is required for the innate immune response to EMCV vRNA.**
**A**–**C** WT and *Adam9* KO primary lung fibroblasts were transfected with 0.1, 1, or 10 ng of EMCV vRNA or mock-transfected (using lipofectamine only, Lipo, as a control), and 24 h later, EMCV titers in the supernatant were determined by plaque assay (**A**) and IFN-β (**B**) or IL-6 (**C**) measured by ELISA. **D** IFN-β produced by WT and *Adam9* KO lung fibroblasts infected with different levels of Sendai virus (SeV) as measured using hemagglutinin (HA) units for 24 h, measured by ELISA. **E** *Ifnb1* mRNA in WT and *Adam9* KO mouse lung fibroblasts transfected with the RIG-I ligand RABV-Le RNA. **F** IFN-β levels from VSV vRNA-transfected lung fibroblasts, measured by ELISA. **G** IFN-β levels measured in supernatants from WT, *Adam9* KO, and ADAM9-rescued fibroblasts at 24 h post-transfection with EMCV vRNA, determined by ELISA. In **A**–**D** and **F** one representative experiment of three independent experiments with $n = 2$ in each group is shown. In **E** and **G** data are representative of two independent experiments with $n = 3$ (**E**) or $n = 2$ (**G**) in each group. Error bars show mean ± SD. In **B**, **C**, and **G** *P* values were determined by two-way ANOVA with Tukey's multiple comparisons test; ****$P < 0.0001$; statistically not significant, ns, $P = 0.8954$, ***$P = 0.0005$, ****$P < 0.0001$; **$P = 0.0023$, ****$P < 0.0001$. Source data are provided in the Source Data file.

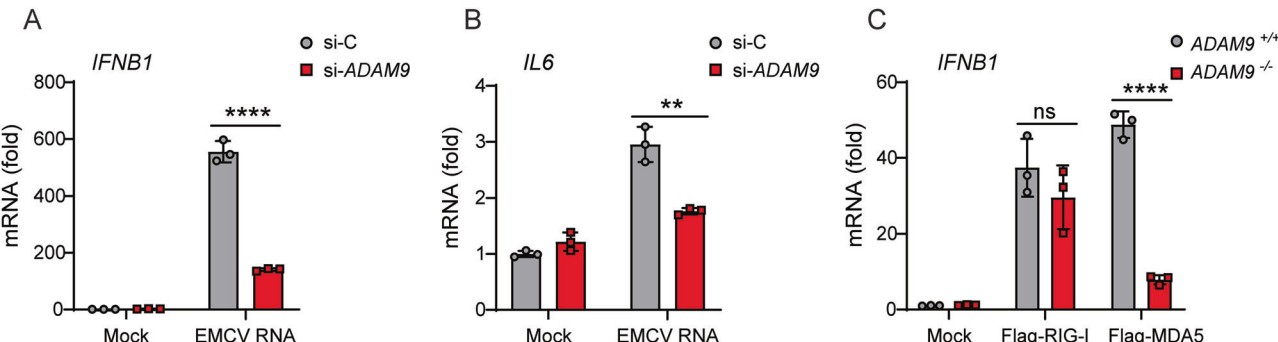

**Fig. 7 | ADAM9 mediates MDA5-induced IFN-β and IL-6 responses to EMCV RNA.**
**A**, **B** Primary normal human lung fibroblasts (NHLFs) were transfected with non-targeting control siRNA (si-C) or *ADAM9*-specific siRNA (si-*ADAM9*) for 30 h and then transfected with EMCV RNA (100 ng ml⁻¹) for 8 h. RT-qPCR analysis of *IFNB1* (**A**) and *IL-6* (**B**) transcripts in control and *ADAM9*-silenced cells. **C** RT-qPCR analysis of *IFNB1* transcripts in *ADAM9*⁺/⁺ and *ADAM9*⁻/⁻ HeLa cells that were transfected with vector (mock), FLAG-hRIG-I, or FLAG-hMDA5 for 24 h. Data are representative of two independent experiments. Error bars are mean ± SD of $n = 3$ replicates. ****$P < 0.0001$; **$P = 0.003$; ns ($P = 0.2997$), statistically not significant, (two-tailed, unpaired *t*-test). Source data are provided in the Source Data file.

with this, mADAM9 expression induced MAVS aggregation, which is indicative of MAVS activation. Notably, mADAM9 E→A mutant exhibited a similar effect as seen for WT mADAM9 on MDA5 oligomerization and MAVS aggregation (Fig. 8E), further strengthening our assertion that ADAM9's catalytic activity is dispensable for potentiating MDA5 signal activation. Collectively, these findings establish that ADAM9, independent of its catalytic activity, mediates vRNA-induced type I IFN production by facilitating MDA5 oligomerization and downstream MAVS activation.

## Discussion

In the present study, we used a well-established murine model of viral myocarditis to determine the role of ADAM9 in EMCV infection and subsequent development of virus-induced cardiac injury. EMCV infection was associated with significantly increased virus replication in the heart and serum of *Adam9* KO mice compared to WT mice. In fact, EMCV-infected *Adam9* KO mice demonstrated massive cardiac damage compared to WT mice. *Adam9* KO mice were exquisitely sensitive to EMCV infection with 100% mortality within 2–3 days with

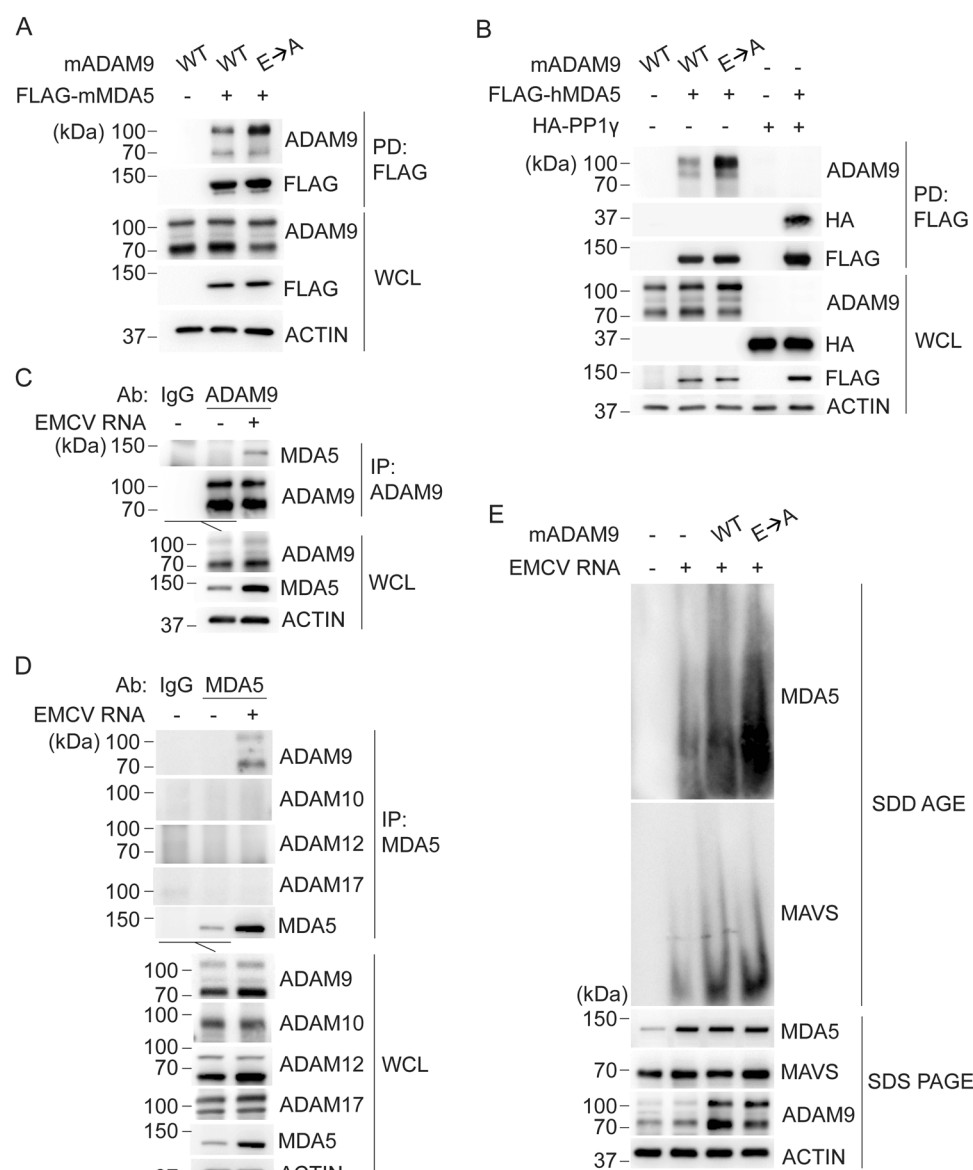

**Fig. 8 | ADAM9 binds MDA5 promoting MDA5 oligomerization and downstream MAVS activation. A** Binding of FLAG-tagged murine MDA5 (FLAG-mMDA5) to murine ADAM9 (mADAM9) WT or catalytically inactive mutant (E→A) that were co-expressed for 24 h in HEK293T cells, determined by FLAG pull-down (PD: FLAG) and IB with the indicated antibodies. **B** Binding of FLAG-tagged human MDA5 (FLAG-hMDA5) to mADAM9 WT or E→A mutant, or HA-PP1γ (positive control[26]), that were co-expressed for 24 h in HEK293T cells, determined by PD:FLAG and IB with the indicated antibodies. **C** Binding of endogenous ADAM9 to MDA5 in LFs that were either mock-treated or transfected with EMCV RNA (400 ng ml⁻¹) for 16 h, determined by IP with anti-ADAM9 (or an IgG isotype control) and IB with the indicated antibodies. **D** Binding of endogenous ADAM9, ADAM10, ADAM12, or ADAM17 to MDA5 in LFs that were either mock-treated or transfected with EMCV RNA (400 ng ml⁻¹) for 16 h, determined by IP with anti-MDA5 (or an IgG isotype control) and IB with the indicated antibodies. **E** Endogenous MDA5 oligomerization and MAVS aggregation in Hepa1-6 cells transfected for 24 h with vector or mADAM9 WT or E→A and then transfected for 16 h with EMCV RNA (400 ng ml⁻¹), assessed by SDD-AGE and IB with anti-MDA5 and anti-MAVS. Whole cell lysates were further analyzed by SDS-PAGE and probed by IB with the indicated antibodies. Data are representative of at least two independent experiments. Source data are provided in the Source Data file.

accelerated kinetics compared to WT mice. EMCV-infected *Adam9* KO mice also exhibited significantly higher serum cTNI levels than infected WT mice at 48 h p.i., indicating cardiomyocyte damage. Immunohistochemistry confirmed structural damage with corresponding troponin leakage in the hearts of *Adam9* KO mice following EMCV infection. These cardiac changes were not seen in infected WT mice. Thus, ADAM9 appears to protect from EMCV-induced cardiac morbidity and mortality in mice. Altogether, our in vivo data demonstrate that ADAM9 plays a previously unknown cardioprotective role during EMCV infection.

Caforio et al.[30] defined myocarditis histologically as an inflammatory process of myocardial tissues. Our results are not entirely

consistent with that definition as we did not see evidence of inflammatory infiltrates in the hearts of infected *Adam9* KO mice. However, histologic evidence of cardiac damage was apparent in *Adam9* KO mice in the absence of inflammatory cell infiltration. In the case of EMCV-induced cardiac death (and possibly other viruses that cause rapid cellular damage), the term "myocarditis" may be a misnomer, as the myocyte damage is caused directly by the virus in the absence of inflammatory infiltrates. These data are compatible with the results of Kato et al., who noted that *Mda5* KO mice infected with EMCV developed "virus-induced cardiomyopathy"[27] and also the results of Philip et al., who found that selectively expressing MDA5 in mouse cardiomyocytes prevented EMCV-induced cardiac dysfunction[31]. In contrast

to EMCV, mice infected with Coxsackievirus developed myocarditis associated with host immune responses and characterized by an inflammatory cell infiltrate in the myocardium as described by others[32]. In our study, post-mortem histology revealed that infected *Adam9* KO mice develop a severe dilated cardiomyopathy along with increased viral proliferation compared to WT controls. Our results suggest that, in contrast to WT controls, infected *Adam9* KO mice develop fulminant myocardial damage heralded by architectural distortion of cardiac tissue (i.e. intramyofiber disruption) and chamber dilation, which can lead to severe functional impairment and conduction abnormalities ultimately triggering fatal arrhythmias, which we postulate to be the cause of death in these animals. Thus, we hypothesize that the accelerated mortality of infected *Adam9* KO mice is due to sudden cardiac death precipitated by direct virus-induced cardiac damage in the absence of a type I IFN response, further supporting a cardioprotective role for ADAM9 during EMCV infection in vivo.

Type I IFN is critical to protect the heart from EMCV-induced damage. Studies of mice that are deficient in the type I IFN receptor (IFNAR) demonstrate that IFNAR is essential for protection from EMCV infection[33]. IFNAR engagement by IFN-β and other type I IFNs drives paracrine signaling events resulting in the expression of hundreds of antiviral IFN-stimulated genes (ISGs). Interestingly, MDA5, the major sensor for EMCV in mice, is itself an ISG. Another ISG, IFIT1, is an essential antiviral gene that blocks picornavirus replication[34]. Studies in mice with selective knockout of IFNAR in cardiac cells suggest that in the absence of type I IFN signals, MDA5 drives a hyper-inflammatory cytokine response in the heart rather than a protective antiviral response[35]; however, we did not detect inflammatory cytokines in either the heart or serum of infected *Adam9* KOs, indicating that, despite high levels of virus in these tissues, MDA5 was not activated in the absence of ADAM9.

We have also previously demonstrated a critical role for type I IFN in protecting mice from lethal picornavirus infection. Wang et al. demonstrated that injection of BALB/c mice with recombinant mouse IFN-β protects mice from otherwise lethal CVB3 challenge[36]. In the present study, we discovered that *Adam9* KO mice have elevated levels of EMCV in the heart and lower levels of IFN-β in the heart and serum, suggesting that the failure to produce type I IFN may be driving the accelerated mortality observed in EMCV-infected *Adam9* KO mice.

ADAM9 is a binding/entry receptor for EMCV on murine fibroblasts[7]; however, it is not required for EMCV infection of cardiac tissue nor is it required for EMCV replication post-entry. By transfecting vRNA to bypass receptor-mediated cell entry, we confirmed that ADAM9 is not required for EMCV replication. Nevertheless, ADAM9 is required for type I IFN production in EMCV vRNA-transfected cells. IFN-β responses are abrogated in *Adam9* KO cells transfected with vRNA, despite high levels of replicating EMCV in these cells. This result suggests a link between ADAM9 and intracellular recognition of cytosolic vRNA by innate immune sensors. The lack of antiviral immunity likely contributed to the enhanced EMCV replication in *Adam9* KO heart tissues and the consequent damage to the myocardium, as evidenced by high levels of serum troponin and disruption of myocardial structures in *Adam9* KO mice infected with EMCV. Together, these data reveal a previously unrecognized role for ADAM9 in host defense against picornaviruses by regulating the type I IFN response.

MDA5 is a sensor of double-stranded RNA (dsRNA) and is the major innate immune receptor for picornaviruses, including EMCV and CVB3. Experiments by Kato et al.[27] indicate that *Mda5* KO mice rapidly succumb to EMCV infection. Similarly, Gitlin et al.[4] demonstrated that *Mda5* KO mice are particularly sensitive to low doses of EMCV. Activation of MDA5 occurs when dsRNA replication intermediates are loaded onto MDA5[26,37]. MDA5-vRNA complexes form filaments and bind to MAVS, initiating downstream signaling for IFN/cytokine gene

expression via TBK-1 and IKKα/β/γ activation. EMCV triggers MDA5-MAVS signaling that drives the phosphorylation and activation of IRF3 and NF-κB, ultimately eliciting type I IFN gene expression[5,27,38–41]. In addition to driving type I IFN production, activation of the MDA5-MAVS axis by EMCV generates innate inflammatory cytokines and chemokines, including IL-6[4,27,42]. Thus, during EMCV infection, serum IFN-β and IL-6 responses are absent in *Mda5* and *Mavs* KO mice, consistent with our observations in *Adam9* KO mice.

MDA5 has been recognized as a sensor for SARS-CoV-2 infection in several cell types[43,44]. MDA5 is also activated by immunization with the COVID-19 mRNA vaccine, where MDA5 plays a critical role in the development of T cell immunity to the vaccine[45]. The involvement of MDA5 as a sensor for dsRNA and replication intermediates of positive-strand RNA viruses as well as the Pfizer-BioNTech BNT162b2 mRNA vaccine points to the MDA5 pathway as a core innate immune response to infection and vaccination[45]. By extension, ADAM9 may also play a central role in the innate immune response to positive-strand RNA viruses by interacting with MDA5 and thus supporting both innate type I IFN and inflammatory cytokine responses and subsequently promoting the development of adaptive antiviral immunity.

In contrast to EMCV vRNA, challenge with either Sendai virus, RABV-Le RNA, or VSV vRNA, all of which induce type I IFN responses via RIG-I, produced a robust IFN-β response from both WT and *Adam9* KO cells. Indeed, our data showed that ADAM9 deficiency markedly reduced *IFNB1* and *IL-6* mRNA induction mediated by ectopic MDA5, but not RIG-I, indicating that ADAM9 regulates the innate antiviral response via an MDA5-dependent signaling pathway. MDA5 activation requires RNA binding, oligomerization, and translocation from the cytosol to mitochondria for interaction with MAVS[37]. Our findings reveal that ADAM9 binds to MDA5, promoting its higher-oligomer formation and downstream MAVS activation. Together, these results suggest that ADAM9 affects the type I IFN response upstream of MAVS at the level of vRNA-MDA5 interaction or vRNA-mediated MDA5 oligomerization, as indicated by our biochemical data. It is also possible that ADAM9 affects further upstream activation steps of MDA5, including certain post-translational modifications known to modulate MDA5 activity (e.g., PP1-PPP1R12C-mediated dephosphorylation, ISGylation, and K63-linked ubiquitination)[26,44,46,47], which warrants further investigation. Moreover, the unique molecular and/or structural features underlying the interaction of ADAM9 with MDA5 remain to be investigated, given that other ADAMs (i.e., ADAM10, ADAM12, and ADAM17) did not bind to MDA5 in response to EMCV vRNA stimulation. Notably, the catalytic activity of ADAM9 appears to be dispensable for its binding to MDA5 and facilitation of MDA5 oligomerization, suggesting an unconventional function of ADAM9 in regulating MDA5 activation.

The mechanism by which cell surface proteins such as ADAM9 regulate MDA5 signaling is currently unknown. The cytosolic tail of ADAM9 contains SH3 domains that might directly impact MDA5 signaling[19–21]. Alternatively, ADAM9 could bind to and/or affect MDA5 via interaction with other cellular proteins, such as integrins. ADAM9 associates with β1-integrins to regulate cell migration[13,14], and our recent work revealed a link between actin filament disruption and RLR signaling, whereby cytoskeletal perturbations promote cytosolic RLR signaling[46]. ADAM9 can also proteolytically process ADAM10, which may influence antiviral immunity[23,48–51]. However, our studies indicate that the proteolytic activity of ADAM9 is dispensable for MDA5 signaling, and that, in contrast to ADAM9, we do not observe an association of ADAM10, ADAM12, or ADAM17 with MDA5.

To our knowledge, our study is the first to identify a direct role for ADAM9 in the innate immune response to viral infection. We demonstrate that ADAM9 is necessary for MDA5 signaling and has a unique role as a cardioprotective factor during EMCV infection. Thus, cardiac cells are infected in ADAM9 KOs but fail to initiate MDA5 signaling for

antiviral IFN-β and innate cytokine production. We hypothesize that ADAM9-expressing cells, including fibroblasts that express high levels of ADAM9, produce type I IFNs via an MDA5-dependent signaling pathway, which then acts on cardiomyocytes via IFNAR to promote antiviral gene expression and protect cardiomyocytes from EMCV infection and limit direct virus-mediated cardiac damage.

# Methods

## Ethics
All experiments performed using animals were approved by the University of Massachusetts Chan Medical School Institutional Animal Care and Use Committee (PROTO202200076 [A-2211]).

## Cell lines
Mouse lung fibroblasts were isolated from WT and *Adam9* KO mice and immortalized with SV40, as previously described[7]. In brief, primary lung fibroblasts were isolated from 6- to 10-week-old C57BL/6J WT and ADAM9 KO mice. Mice were euthanized and then washed with 90% ethanol to reduce contamination and then whole-lung lobes were dissected and placed into sterile PBS. Lungs were then submerged in 70% ethanol for 20 s, placed into digestion medium (DMEM with 0.25% trypsin), and cut into small pieces using a sterile scalpel. The lung pieces were then transferred into 10 ml of digestion medium and incubated with shaking at 37 °C. After 30 min, 5 ml of cold complete DMEM (DMEM, 10% heat-inactivated FBS, 2 mM L-glutamine, 100 U/ml penicillin, 100 μg/ml streptomycin, 250 ng/ml amphotericin B) was added, and cells were centrifuged for 5 min at 350 × *g* at 4 °C. The cell pellet was removed and resuspended in fresh complete DMEM. For each mouse, the lung pieces and cells were resuspended in 50 ml of complete DMEM and then plated in a 225 cm² tissue culture flask. The cells were incubated for 10–15 days with minimal disruption, with fresh medium added after 7 days. For the first passage, the cells and lung pieces were detached from the flasks by scraping and separated with a 70 μm cell strainer. The single-cell suspension was either passaged one or two times when 90% confluent in complete DMEM or used immediately for stimulation. Fibroblasts were immortalized with SV40. HeLa (Catalog # CRM-CCL-2), Vero E6 (Catalog # CRL-1586), BHK-21 (Catalog # CCL-10), HEK293T (Catalog # CRL-3216), and Hepa1-6 (Catalog # CRL-1830) cells were obtained from the American Type Culture Collection (ATCC), which performed quality control tests, including testing for mycoplasma, STR profiling, and quality accreditation.

## Mouse lines
Wild-type C57BL/6J mice were purchased from Jackson Laboratories. The ADAM9 KO mice were originally generated by Dr. Blobel[52] and breeding pairs were provided by Dr. Carolyn Owen. Mice were fully backbred onto the C57BL/6J background and backbreeding was confirmed by satellite testing (Jackson Laboratories). ADAM9 knockout was confirmed by qPCR and WB analysis. All animals were housed with four mice per cage in a room with a 12:12 light/dark cycle at 70 ˚F ± 20 ˚F under 30–70% humidity conditions.

## Virus strains, and viral RNA preparation
EMCV VR129B and VSV strain Indiana were obtained from ATCC, and virus stocks were propagated in Vero E6 cells. Sendai virus (Cantell strain) was purchased from Charles River Laboratories (North Franklin, CT). CVB3 Nancy was obtained from Dr. Finberg. EMCV and VSV vRNA was isolated from ultrapure virus stocks using the QIAamp Viral RNA Mini Kit (Qiagen), according to the manufacturer's protocol. RABV-Le RNA was generated by in vitro transcription using annealed DNA oligos (IDT) as a template and the MEGAshortscript T7 Transcription Kit (Invitrogen) as previously described[53]. Viral RNA concentration was determined by NanoDrop, and vRNA stocks were stored at −20 °C until use. Cells were transfected with vRNA using Lipofectamine2000 (ThermoFisher Scientific, catalog no. 11668027) as previously described[7].

## Plasmid construction
FLAG-RIG-I, FLAG-MDA5 (hMDA5), and HA-PP1γ were previously described[26]. FLAG-mMDA5 was cloned by ligating a synthetic gBlocks gene fragment (IDT) containing the mMDA5 ORF into pcDNA3.1/3xFLAG between *Nhe*I and *Not*I.

## EMCV murine infection
Age-matched WT and *Adam9* KO mice were infected i.p. with EMCV at 10⁵ PFU/mouse in 200 μL of sterile PBS. Mice were euthanized at the indicated times post-infection. Blood was harvested by direct cardiac puncture and placed into a BD Microtainer serum separation tube (BD Biosciences, Franklin Lakes, NJ), centrifuged at 9000 × *g* for 5 min, and then stored at −80 °C. Mouse hearts were harvested, bisected vertically, and one half placed into sterile 1× PBS and the other half into Trizol reagent and kept on ice. Following sample collection, organs were homogenized using Qiagen TissueRuptor (Qiagen.com) at maximum speed for 10 s per sample. Homogenates were clarified by centrifugation at 12,000 × *g* for 10 min at 4 °C, and supernatants were removed and placed into new sterile 1.5 mL tubes and stored at −80 °C.

## EMCV virus titering by plaque assay
BHK-21 cells were plated onto 12-well plates to form a complete monolayer. The samples to be tested were serially diluted and added onto the BHK-21 cells in duplicate with a final volume of 500 μL per well. The plates were incubated at 37 °C for 1 h with intermittent rocking for initial infection. The wells were washed with PBS and overlayed with 2% agar in minimum essential medium, and the overlay was allowed to solidify at room temperature and then incubated at 37 °C for -24–26 h until visible plaques were observed in the cell monolayer. Cells were fixed and stained by adding 4% PFA and 0.5% crystal violet in PBS for at least 2 h. The overlay was gently removed by washing with running tap water, and the plates were allowed to dry on the benchtop. The plaques were enumerated, and PFU/mL was calculated.

## Serum and tissue ELISA and Luminex cytokine assays
Antibodies to detect cardiac troponin I (cTNI, Life diagnostics), IFN-β (PBL Assay Science), and IL-6 (R&D) were used to perform ELISAs according to the manufacturer's instructions. Briefly, Maxisorp 96-well plates were coated with the capture antibodies in appropriate buffers overnight at 4 °C as directed. Plates were blocked with appropriate buffers (1% BSA or 5% FBS in PBS, or no blocking in the case of the IFN-β ELISA). The tissue and serum samples were diluted as needed in blocking buffer along with a standard curve of purified protein. The blocking buffer was removed from the plates, and samples were added to the wells and incubated at 4 °C while shaking. The wells were washed 3× with washing buffer, and the detection antibody was added and incubated at 4 °C while shaking. After washing, a secondary horseradish peroxidase (HRP) conjugate was added and incubated at 4 °C. The wells were washed 5×, and TMB (3,3′,5,5′-tetramethylbenzidine) reagent was used for readout on an ELISA plate reader. Serum cytokines (RANTES, IL-6, MIP-1α, MIP-1β, KC/CXCL1) were measured by Luminex multiplex magnetic bead-based assay using the Bio-Plex Pro™ Mouse Cytokine 23-Plex, Group 1 (Bio-Rad Laboratories, Hercules, CA, Cat# M60009RDPD) according to the manufacturer's instructions. Data were acquired on a calibrated Bio-Rad Bio-Plex MAGPIX Multiplex Reader and analyzed with Bio-Plex Manager software (Bio-Rad). All standards and samples were measured in duplicate.

## Cardiac histology

Histological staining was performed as previously described[54]. Briefly, murine hearts were fixed in 4% paraformaldehyde (PFA) overnight at 4 °C. After fixation, tissues were dehydrated in ethanol, embedded in paraffin, and sectioned at 8 μm thickness using a Leica rotarized microtome. Sections were then deparaffinized in xylene and rehydrated with an ethanol gradient. After rehydration, sections were stained with Harris-modified hematoxylin (2 min) and eosin-Y staining (30 s). Slides were then dehydrated via an ethanol gradient and cleared with xylene. Cleared slides were mounted with Vectashield glass mounting medium.

## Cardiac immunohistochemistry

Immunohistochemistry was performed similarly to previously described[55]. Sectioned adult hearts were deparaffinized, immersed in sodium citrate buffer (10 mM sodium citrate, 0.05% Tween-20, pH 6.0), and placed in a 2100 Antigen Retriever (Aptum Biologics) for antigen retrieval. After antigen retrieval, sections were blocked in 10% donkey serum, 1% BSA, and 0.3% Triton X-100 in PBS for 1 h at room temperature and incubated with primary antibodies against cardiac troponin-T (DSHB, Cat no RV-C2), EMCV, or the membrane stain wheat germ agglutinin at dilutions of 1:50 for troponin, 1:500 for EMCV, and 2 μg/mL wheat germ agglutinin. Following primary incubation, slides were washed with PBS and incubated with Alexa Fluor 488 or 546 conjugated secondary antibodies (1:500) with Hoechst nuclear stain (1:1000) at room temperature for one hour. After secondary incubation, slides were rinsed with PBS and mounted with Prolong Glass Antifade Mountant (Invitrogen, cat. no. P36980).

## Cardiac imaging

Hematoxylin and eosin (H&E)-stained section images were captured with a Nikon Eclipse 80i microscope using a CFI Plan Fluor 4×/10×/20×/40× objective lenses, DS-Fi1 color camera, and NIS-Elements Basic Research software. Immunofluorescent images were captured on a Zeiss LSM800 Airyscan inverted digital spectral confocal microscope equipped with Definite Focus 2.0, three confocal GaAsP detectors, including an Airyscan detector with 6000 × 6000-pixel resolution, and a solid-state laser module with 405-, 488-, 561-, and 640-nm beam splitter. Plan-Apochromat objectives (63×/1.4 oil and 20×/0.8) with DIC were used for image capture, and image stacks of maximal vertical projections were assembled with Zeiss Zen 2.5 imaging software.

## Quantification of myofiber-free space

Extracellular space was quantified using ImageJ software (NIH, Bethesda, MD). H&E images obtained at 20× were converted to a binary image and inverted so that white pixels corresponded to the extracellular space. Regions of interest (ROI) were manually drawn around each myofiber. Then, within each ROI, the number of white versus black pixels was counted. Finally, the percentage of white pixels was calculated per ROI, and they were then pooled to develop a mean and standard deviation per image and treatment condition. The number of ROI, or myofibers, per image was 6–12.

## Preparation of infected cells for RNAscope™

WT and *ADAM9* KO HeLa H1 cells were seeded into chamber slides (8-chamber/slides, 60,000 cells/chamber). The next day, cells were infected with EMCV at an MOI = 10. Virus was diluted in serum-free media and 100 μL was added to each chamber. The cells were incubated at 37 °C with 5% CO$_2$ for one hour with gentle shaking every 15 min. The experiment was performed with triplicate chambers for each infection and uninfected control. The virus media was removed, 400 μL of complete growth media was added to each chamber, and the cells were incubated at 37 °C with 5% CO$_2$. Infected cells were harvested at 2.5 and 5 h p.i. by removing growth media, washing with PBS once, and then fixing with 4% PFA for 30 min at RT. After fixing, cells

were washed with PBS 3×, dehydrated with 50% ethanol for 1 min, 70% ethanol for 1 min, and then 100% ethanol for 1 min. Slides were stored in 100% ethanol at −20 °C prior to staining.

Cells were rehydrated according to ACDBio RNAscope™ Multiplex fluorescent Reagent Kit v2 Assay protocol (Advanced Cell Diagnostics, Inc., Newark, CA). Briefly, slides were placed in 100% ethanol for 1 min, 70% ethanol for 1 min, 50% ethanol for 1 min, and then PBS for 1 min. Cells were permeabilized in 1× PBS + 0.1% Tween-20 for 10 min at RT and then treated with protease III for 10 min at RT. The RNAscope™ assay was performed following the manufacturer's protocol.

## Preparation of heart sections for RNAscope™

Mouse hearts were harvested 24 h p.i. with EMCV. Samples were fixed with 4% PFA and then transferred to ethanol before paraffin embedding. Samples were then cut into 8-micron sections on regular slides and stored at 4 °C. The sections were baked and deparaffinized following the manual of Multiplex Fluorescent Reagent Kit v2 Assay from ACD. Target retrieval was performed for 15 min using a steamer (Hamilton Beach Food Steamer model 37530A).

## RNAscope™ probes

RNAscope probes were synthesized by ACDBio (Advanced Cell Diagnostics, Inc., Newark, CA). For the EMCV probe, we used the 20zz pair anti-sense probe (V-EMCV-VP targeting 1856-2763; GenBank Accession #: KM269482.1) in the C2 channel. We used a 1:50 dilution as the working concentration for the EMCV-specific RNAscope™ probe-C2 mCherry.

As a positive control, we used the species-specific 3-Plex Positive Control Probe, which is a mixture of the following three probes:
- Probe targeting POLR2 in Channel C1−FITC low expression positive control
- Probe targeting PPIB in Channel C2−Cy3 moderate expression positive control
- Probe targeting UBC in Channel C3−Cy5 high expression positive control

UBC had the highest relative expression levels, while PPIB was moderate-to-high and POLR2A was moderate-to-low. For a negative control, we used the 3-Plex Negative Control Probe that targets a soil bacterial gene *dapB* in all three channels; thus, each detection channel has its own negative control probe.

## RNAscope™ slide preparation, detection, and analysis

The slides were mounted following the ACDBio RNAscope™ protocol after developing the HRP-C2 signal (for the EMCV-specific RNAscope™ probe-C2 mCherry). DAPI was used as a counterstain. Briefly, after washing the slides, excess liquid was removed, and ~4 drops of DAPI were added to each slide and incubated for 30 s at RT, and then the DAPI was removed by gently tapping the slides. Immediately after, Prolong Gold Antifade Mountant (1–2 drops, Invitrogen) was added to each slide, and a 24 × 50 mm glass coverslip was placed over the tissue section before drying the slides overnight in the dark. Slides were stored in the dark at 2–8 °C.

The EMCV-specific RNAscope™ probe-C2 was used at a 1:50 dilution and detected with Opal 570 dye (Akoya Biosciences). The EMCV genome was visualized with fluorescence microscopy in the Cy3 filter. Both the EMCV and control probes were detectable in one of the three fluorescent channels (C1, C2, and C3). Fluorescence signals for C1, C2, and C3 channels were generated using Opal dyes (Akoya BioSciences) 520 (FITC filter), 570 (Cy3 filter), and 690 (Cy5 filter), respectively. EMCV RNA genomes were visualized by fluorescence microscopy in the UMass Chan Sanderson Center for Optical Experimentation (SCOPE) core facility (SCOPE RRID SCR_022721). Images of RNAscope™ stained cells and heart sections were

## Table 1 | Details on PrimeTime qPCR assay primers

| Name | Source | Identifier |
|---|---|---|
| Human IFNB1 | Integrated DNA Technologies | Hs.PT.58.39481063.g |
| Human GAPDH | Integrated DNA Technologies | Hs.PT.39a.22214836 |
| Human IL-6 | Integrated DNA Technologies | Hs.PT.58.40226675 |

acquired with a TissueFAXS PLUS v6 microscope (TissueGnostics GmbH) built on a Zeiss AxioImager.Z2 microscope base with Märzhäuser stage, Lumencor Spectra X light engine with 390/22 nm and 575/22 nm excitation filters, Chroma 89402 and 89403 quad emission filters, Zeiss EC Plan-Neofluar 20×/0.5 NA objective and Hamamatsu Orca Flash 4 camera. Data were analyzed using Strata-Quest Plus V6.0 software (TissueGnostics GmbH). Nuclei were detected in the DAPI channel, and cell masks were generated by expanding nuclei boundaries. The cellular intensity in the Texas Red channel was quantified, and cutoffs were determined for positive cells using unstained negative controls. Positive cells were quantified as a percentage of the total cell population for each sample.

### RT-qPCR analysis
Isolation of RNA and RT-qPCR analysis were performed as described previously[46]. Briefly, total RNA was extracted from indicated cells using the E.Z.N.A. HP Total RNA Kit (Omega Bio-tek) per the manufacturer's instructions. Equal RNA amounts of total RNA were used for qRT-PCR analysis using the SuperScript III Platinum One-Step qRT-PCR kit with ROX and predesigned PrimeTime qPCR Probe Assays (IDT, see Table 1) on a QuantStudio 6 (Applied Biosystems). mRNA expression was normalized to the levels of *GAPDH* and expressed relative to the values for control cells using the ΔΔCt method.

### siRNA-mediated ADAM9 knockdown
Transient gene knockdown experiments were performed using gene-specific siGENOME SMARTpool small interfering RNAs (Horizon Discovery) as described previously[56]. Specifically, for ADAM9 knockdown in NHLF cells, non-targeting siRNA Pool no. 2 (D-001206-14) (control) and *ADAM9* siRNA SMARTpool (M-004504-03) were used. Lipofectamine RNAiMAX Transfection Reagent (Invitrogen) was used as per the manufacturer's instructions for siRNA transfection.

### Rescue experiments
ADAM9 rescue of the knockout cell lines was done by transduction as previously described[7]. Retrovirus was made using 293FT cells transfected with retroviral transfer constructs with a pQCXIP backbone (GFP, Addgene catalog no. 73014; mouse Adam9 WT; mouse Adam9 E > A), pMD 2.G, and VSV-G packaging plasmid using TransIT-293 transfection reagent (Mirus 2704) per protocol[57]. After 24 h, the medium was replaced with complete DMEM. At 24 h and 48 h, the supernatant was collected (and replaced with complete DMEM as needed) and filtered through a 0.45 μm filter. Virus supernatants were used immediately or frozen at −80 °C in single-use aliquots.

### EMCV RNA purification
EMCV RNA was produced as previously described[44]. Briefly, Vero cells (CCL-81, ATCC) were infected for 16 h with EMCV (MOI = 0.1), and total RNA was isolated using the Direct-zol RNA extraction kit (Zymo Research) following the manufacturer's instructions.

### SDD-AGE
MDA5 oligomerization induced by EMCV RNA transfection was determined by SDD-AGE as previously described[44]. MAVS aggregation was assessed using a published protocol[58] with slight modifications. Briefly, LFs were transfected with EMCV RNA (400 ng ml$^{-1}$) for 16 h. Cells were harvested and resuspended in buffer A (10 mM Tris-HCl [pH 7.5], 1.5 mM MgCl$_2$, 10 mM KCl, 0.25 M D-mannitol) and then lysed by grinding. Cell debris and the supernatant were separated by centrifugation (at 700 × g for 10 min). The supernatant was then centrifuged at 10,000 × g for 30 min at 4 °C in order to separate crude mitochondria and cytosolic extracts. The crude mitochondria extract was resuspended in 1× sample buffer (0.5 × TBE, 10% (v/v) glycerol, 2% (w/v) SDS, and 0.0025% (w/v) bromophenol blue) and loaded onto a 1.5% agarose gel. Samples were subjected to electrophoresis and IB analysis using the indicated antibodies.

### Immunoprecipitation (IP) and immunoblotting (IB)
Immunoprecipitation of ectopically or endogenously expressed proteins was performed as previously described[46,59]. Briefly, cells were lysed at the indicated times in NP-40 buffer (50 mM HEPES [pH 7.4], 150 mM NaCl, 1% (v/v) NP-40, 1 mM EDTA) or radio-immunoprecipitation assay (RIPA) buffer (50 mM Tris-HCl [pH 7.4], 150 mM NaCl, 1% (v/v) NP-40, 0.5% (w/v) deoxycholic acid, 0.1% (w/v) SDS) with protease inhibitor cocktail (Sigma–Aldrich, Cat# P2714). Lysates were subsequently cleared of cellular debris via centrifugation at 21,000 × g for 20 min at 4 °C. A portion of the cleared lysate was analyzed by IB for protein expressions in the whole cell lysate (WCL), while the remaining lysate was subjected to co-immunoprecipitation. For FLAG pulldowns, lysates were incubated at 4 °C with anti-FLAG M2 magnetic beads for 4–16 h. For co-immunoprecipitation of endogenous proteins, cell lysates were incubated overnight at 4 °C with approximately 1–2 mg ml$^{-1}$ of the respective primary antibody, followed by incubation with protein G-conjugated magnetic beads for an additional 2 h at 4 °C.

### Antibodies and reagents
Anti-ADAM9 (CST, Cat# 2099S, 1:100) for IP; anti-ADAM9 (Bio-Techne Corporation, Cat# AF949, 1:1,000) for IB; anti-ADAM10 (Proteintech, Cat# 25900-1-AP, 1:1,000); anti-ADAM12 (Proteintech, Cat# 14139-1-AP, 1:1,000); anti-ADAM17 (Proteintech, Cat# 29948-1-AP, 1:1,000); anti-MDA5 (CST, Cat# 5321, 1:1,000); anti-MAVS (Santa Cruz, Cat# sc-365333, C1, 1:500); anti-HA (CST, Cat# 3724, 1:1,000); anti-ACTIN (GeneTex, Cat# GTX629630, GT5512, 1:2,000); Normal Rabbit IgG (CST, Cat# 2729, 1 mg ml$^{-1}$); Anti-TNNT (DSHB, Cat# RV-C2, 1:50); PEI (Polysciences, Cat# 00618); Lipofectamine2000 (ThermoFisher, Cat# 11668019); Dynabeads Protein G (Invitrogen, Cat# 10009D); anti-FLAG M2 beads (Sigma–Aldrich, Cat# M8823); protease inhibitor cocktail (Sigma–Aldrich, Cat# P2714).

### Statistical analysis
GraphPad Prism software (version 9.0; GraphPad, San Diego, CA) was used for data analysis using two-tailed, unpaired Student's *t*-test or two-way ANOVA with Tukey's multiple comparison test. For the mouse survival study, Kaplan-Meier survival curves were generated and analyzed for statistical significance. A *P*-value of < 0.05 was considered statistically significant.

### Biological materials
All unique materials used are readily available from the authors or from standard commercial sources.

### Reporting summary
Further information on research design is available in the Nature Portfolio Reporting Summary linked to this article.

## Data availability
The EMCV probe was based on the GenBank Accession #: KM269482.1 Source data are provided with this paper.

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

## Acknowledgements

The authors dedicate this manuscript to the late Dr. Robert Finberg. We thank Carl Blobel for generously providing *Adam9* KO mice and for his advice and expertise in ADAM protein biology. We thank Melanie Trombly for reviewing the manuscript and for excellent editorial assistance. We thank TissueGnostics GmbH for their technical assistance and for the long-term demo of their microscope and analysis software. The work was supported by an NIH NIAID grant R21 AI174534 to E.A.K.-J. and M.U.G.; NHLBI 5T32HL120823-04 to T.C.; an American Heart Association-Myocarditis Foundation (18POST34030152) to L.E.B.; an NIH NIAID (T32 AI095213) to C.R.M.; an NIH NHLBI RO1 HL118100 and HL141377 to C.M.T. and in part by an NIH NIAID R37 AI087846 to M.U.G.

## Author contributions

Conceptualization: E.A.K.-J., R.W.F. and L.E.B.; methodology: E.A.K.-J., R.W.F., J.R.-O., C.A.O., A.L.B., W.M.M., Z.G., C.E.B. and L.E.B.; formal analysis: E.A.K.-J., L.E.B., T.C., C.E.B.; investigation: L.E.B., J.Z., M.K., GQ.L., T.C., C.M.T.; resources: L.E.B., M.K., C.M.T., T.C., M.U.G., C.R.M., P.P.K. and N.Q.; writing–original draft: E.A.K.-J. and L.E.B.; writing–review & editing: E.A.K.-J., L.E.B., C.M.T., R.W.F., M.U.G.; visualization: E.A.K.-J. and L.E.B.; supervision: E.A.K.-J., C.M.T., M.U.G., R.W.F., C.A.O., A.L.B.; project administration: E.A.K.-J. and L.E.B.; funding acquisition: L.E.B., C.M.T., M.U.G., C.R.M., T.C. and R.W.F. All authors have read and agreed to the published version of the manuscript.

## Competing interests

The authors declare no competing interests.
