## [Peer Review File · Nature Communications]

ADAM9 promotes type I interferon-mediated innate immunity during encephalomyocarditis virus infectionREVIEWER COMMENTS

Reviewer #1 (Remarks to the Author):

Summary:

ADAM9, a member of "A Disintegrin And Metalloprotease" family, is a catalytically active membrane-anchored metalloprotease. It has been associated with abnormal growth of blood vessels in the retina and the advancement of tumors. ADAM9 exhibits constitutive catalytic activity in both biochemical and cell-based assays. Moreover, it can cleave multiple membrane proteins, such as EGF, APP, and CD40 *in vitro*.

The present manuscript is an extension of the authors' earlier work and assesses the effect of ADAM9 deficiency on type I interferon-mediated responses during encephalomyocarditis virus (EMCV) infection in an *in vivo* mouse model. The authors show that mice missing ADAM9 are more susceptible to EMCV-induced death and fail to mount the characteristic interferon response proving that there is an essential role for ADAM9 in disease development.

Major points:

This is a straightforward set of experiments with little to criticize. However, a major concern relates to the novelty of some of the findings. As a role for ADAM9 in the early stages of EMCV infection in both human and murine infection was already known, it is perhaps not surprising that ADAM9 also has a role in EMCV-infected cardiac cells.

Another significant issue is that the study is largely based on congenital global loss of ADAM9 characterization, complicating the analyses of the phenotype. Several cell types such as cardiomyocytes, fibroblasts, and diverse immune cell populations have been implicated in disease. While a previous study suggested that ADAM9 functions as a factor promoting EMCV entry, this manuscript supports the idea that ADAM9 is also critical for cardioprotective responses during infection. The exploration of these potentially cell- and time-specific as well as mechanistically divergent responses of ADAM9 could possibly have provided new insights into the mechanism of ADAM9 action during disease. Several conditional and inducible Cre mouse models for cardiomyocyte as well as fibroblast targeting in the healthy and infected heart are available and would allow for more mechanistically-focused studies.

To this reviewer, this manuscript does not add enough to our understanding of ADAM9 to warrant publication in Nature Communications. The authors allude to the question of how ADAM9 might mediate IFN production, but this is never experimentally addressed. Neither is the mechanism of action of ADAM9 considered. Some insight into these issues might have raised the interest level of this work for the general readership of the Nature Communications.

Experimental:

No animal/breeding information is provided making the interpretation of some of the *in vivo* studies rather difficult.

Minor points:

The authors report that inflammatory cell infiltrates were not observed in either wildtype or ADAM9 KO cardiac tissues following EMCV infection. However, in the discussion they conclude that because of the lack of inflammatory infiltrates in the hearts of infected ADAM9 KOs, the early death of ADAM9 KO mice versus wildtype may be due to sudden cardiac death as a result of altered cardiac tissue architecture and fatal arrhythmias. Though, the lack of inflammatory infiltrates in the hearts of infected animals didn't appear to be genotype specific (e.g., wild type animals also died presumably in the absence of inflammatory infiltrates and for similar reasons only 12-24 hours later). In addition, since EMCV usually induces acute myocarditis with sudden death, and myocarditis is generally characterized by cardiac inflammation, the lack of inflammation could be model-specific, raising additional questions about potential functions of ADAM9 on immune cells.

Results in Fig. 2 A and B should be described in the "results section" as well.

Reviewer #2 (Remarks to the Author):

Kurt-Jones and colleagues report on the important function of the metalloprotease ADAM9 for viral (EMCV)-induced encephalomyocarditis, demonstrated in mouse models with different genetic backgrounds and further functional studies in cell culture.

The observation that the lack of ADAM9 increases the susceptibility of mice for encephalomyocarditis virus-induced death was rather unexpected due to previous data from the same group. In 2019, Bazzone et al. demonstrated in an unbiased cell-based CRISPR/Cas9 screen that ADAM9 is a major susceptibility factor in the early stages of encephalomyocarditis virus infection serving as an entry receptor. Hence, the finding in the current manuscript seems to be rather contrary to the previous observation. However, the authors nicely demonstrated that mice deficient for ADAM9 and infected with EMCV drastically fail to promote type I IFN response, which is necessary to prevent viral replication in cardiac cells. Here, MDA5 could be identified as a likely regulatory candidate connecting ADAM9 and the innate immunity pathway. This implicates that ADAM9 can have dual function during EMCV infection dependent on the target cell, either acting as entry receptor or as inducer of IFN response.

The paper provides important new information and merits publication in Nature communications. However, several issues should be addressed prior publication.

1. The responsible molecular pathway, how ADAM9 promotes type I IFN-response has not been sufficiently addressed. If MDA5 is directly interacting with the cytosolic part of ADAM9 should be demonstrated in appropriate biochemical approaches, e.g. by co-immunoprecipitation or any other kind of proximity validation assays.
2. ADAM9 is a proteolytic enzyme and cleavage of specific substrates may be involved to protect from EMCV infection. Although, unbiased substrate identification is rather complicated and probably beyond the scope of this work, the authors should at least use a catalytically inactive variant of ADAM9 to perform cell culture experiments to test if the catalytic activity is of relevance.
3. ADAM9 is known to cleave and solubilize the protease ADAM10, which has also been linked to certain viral infections. This issue should at least be discussed.
4. Fig. 1B,C: What might be the reason that the viral titer decreases in ADAM9 knock-out mice 6 hours post-infection?

Reviewer #3 (Remarks to the Author):

It's a high quality and excellent manuscript. There are some minor questions:

1. Keywords section: as there was no experiment about mitochondrial antiviral-signaling protein (MAVS), so there is no need to include MAVS in keywords.
2. Line 227, the format of ' β ' should be checked and revised.
3. In the description of the results, it is easier for the reader to understand to write clearly figure A or B. e.g. 'ADAM9 mediates innate immune responses to picornavirus infection' section, only indicated as Figure 4 and Figure 5, each set of pictures has several small pictures, so it is easier for readers to understand when listed as Figure 4A, B, C or D clearly.

Point-by-point response

REVIEWER COMMENTS

Reviewer #1 (Remarks to the Author):

Summary:

ADAM9, a member of “A Disintegrin And Metalloprotease” family, is a catalytically active membrane-anchored metalloprotease. It has been associated with abnormal growth of blood vessels in the retina and the advancement of tumors. ADAM9 exhibits constitutive catalytic activity in both biochemical and cell-based assays. Moreover, it can cleave multiple membrane proteins, such as EGF, APP, and CD40 in vitro.

The present manuscript is an extension of the authors’ earlier work and assesses the effect of ADAM9 deficiency on type I interferon-mediated responses during encephalomyocarditis virus (EMCV) infection in an in vivo mouse model. The authors show that mice missing ADAM9 are more susceptible to EMCV-induced death and fail to mount the characteristic interferon response proving that there is an essential role for ADAM9 in disease development.

Major points:

This is a straightforward set of experiments with little to criticize. However, a major concern relates to the novelty of some of the findings. As a role for ADAM9 in the early stages of EMCV infection in both human and murine infection was already known, it is perhaps not surprising that ADAM9 also has a role in EMCV-infected cardiac cells.

We respectfully disagree that this work is not novel or surprising. In fact, based on our previous work in fibroblasts, we predicted that ADAM9 KO mice would be protected from EMCV because their cells would not be infected. Instead, we observed EMCV infection of both the heart and brain in ADAM9 KOs with higher serum and cardiac titers of EMCV in ADAM9 KOs compared to WT, accompanied by damage to the heart and rapid death. Thus we discovered that: 1) ADAM9 is not required for infection of tissues including the heart and brain, i.e., other receptors for EMCV are present in mouse tissues, and 2) ADAM9 is essential for IFN- β and inflammatory cytokine responses during EMCV infection. We have modified the text to clarify these points, since the ADAM9 KOs failed to mount an innate immune response to EMCV despite high levels of infection.

Another significant issue is that the study is largely based on congenital global loss of ADAM9 characterization, complicating the analyses of the phenotype. Several cell types such as cardiomyocytes, fibroblasts, and diverse immune cell populations have been implicated in disease. While a previous study suggested that ADAM9 functions as a factor promoting EMCV entry, this manuscript supports the idea that ADAM9 is also critical for cardioprotective responses during infection. The exploration of these potentially cell- and time-specific as well as mechanistically divergent responses of ADAM9 could possibly have provided new insights into the mechanism of ADAM9 action during disease. Several conditional and inducible Cre mouse models for cardiomyocyte as well as fibroblast targeting in the healthy and infected heart are available and would allow for more mechanistically-focused studies.

We agree that tissue-specific Cre knockout of ADAM9 would deepen our understanding of ADAM9 functions during disease; however, these studies are beyond the scope of the current paper. Instead, we have chosen to focus on understanding how ADAM9 affects MDA5 signaling, as MDA5 is the major innate immune sensor for +strand RNA viruses and MDA5 is known to be critical for the detection of EMCV leading to IFN- β and cytokine responses in vivo.

To this reviewer, this manuscript does not add enough to our understanding of ADAM9 to warrant publication in Nature Communications. The authors allude to the question of how ADAM9 might mediate IFN production, but this is never experimentally addressed. Neither is the mechanism of action of ADAM9 considered. Some insight into these issues might have raised the interest level of this work for the general readership of the Nature Communications.

In this revised manuscript, we provide new mechanistic data (**New Figure 8**) that demonstrate that ADAM9 binds to MDA5 and promotes MDA5 oligomerization leading to downstream activation of MAVS. Thus, ADAM9 has a critical role in the earliest events of innate immune detection of EMCV via interaction with MDA5.

Experimental:

No animal/breeding information is provided making the interpretation of some of the in vivo studies rather difficult.

This information is now included. The ADAM9 KO mice were originally generated by Dr. Blobel, and breeding pairs were provided by Dr. Carolyn Owen. Mice were backbred onto the C57BL/6 background and backbreeding was confirmed by satellite testing (Jackson Laboratories).

Minor points:

The authors report that inflammatory cell infiltrates were not observed in either wildtype or ADAM9 KO cardiac tissues following EMCV infection. However, in the discussion they conclude that because of the lack of inflammatory infiltrates in the hearts of infected ADAM9 KOs, the early death of ADAM9 KO mice versus wildtype may be due to sudden cardiac death as a result of altered cardiac tissue architecture and fatal arrhythmias. Though, the lack of inflammatory infiltrates in the hearts of infected animals didn't appear to be genotype specific (e.g., wild type animals also died presumably in the absence of inflammatory infiltrates and for similar reasons only 12-24 hours later). In addition, since EMCV usually induces acute myocarditis with sudden death, and myocarditis is generally characterized by cardiac inflammation, the lack of inflammation could be model-specific, raising additional questions about potential functions of ADAM9 on immune cells.

We based our suggestion that ADAM9 KOs died of cardiac disease on the release of high levels of cardiac troponin into the circulation as well as altered cardiac architecture in the ADAM9 KOs. These changes were NOT seen in WT mice. In fact, WT mice exhibited signs of neurologic disease prior to death, which were absent in the ADAM9 KOs. Therefore, the cause of death in WT animals may be primarily due to brain infection rather than cardiac disease. This will require further work and identification of the specific brain and heart receptors for EMCV.

Results in Fig. 2 A and B should be described in the "results section" as well.

Thank you for pointing this out. We have now described these results in the Results section.

Reviewer #2 (Remarks to the Author):

Kurt-Jones and colleagues report on the important function of the metalloprotease ADAM9 for viral (EMCV)-induced encephalomyocarditis, demonstrated in mouse models with different genetic backgrounds and further functional studies in cell culture.

The observation that the lack of ADAM9 increases the susceptibility of mice for encephalomyocarditis virus-induced death was rather unexpected due to previous data from the same group. In 2019, Bazzone et al. demonstrated in an unbiased cell-based CRISPR/Cas9 screen that ADAM9 is a major susceptibility factor in the early stages of encephalomyocarditis virus infection serving as an entry receptor. Hence, the finding in the current manuscript seems to be rather contrary to the previous observation. However, the authors nicely demonstrated that mice deficient for ADAM9 and infected with EMCV drastically fail to promote type I IFN response, which is necessary to prevent viral replication in cardiac cells. Here, MDA5 could be identified as a likely regulatory candidate connecting ADAM9 and the innate immunity pathway. This implicates that ADAM9 can have dual function during EMCV infection dependent on the target cell, either acting as entry receptor or as inducer of IFN response.

The paper provides important new information and merits publication in Nature communications. However, several issues should be addressed prior publication.

1. The responsible molecular pathway, how ADAM9 promotes type I IFN-response has not been sufficiently addressed. If

MDA5 is directly interacting with the cytosolic part of ADAM9 should be demonstrated in appropriate biochemical approaches, e.g. by co-immunoprecipitation or any other kind of proximity validation assays.

We have now performed extensive additional biochemical studies (**New Figure 8**) demonstrating that ADAM9 interacts with MDA5 and promotes its oligomerization, which is a hallmark of MDA5 activation. Our new data also show that ADAM9 induces oligomerization of the RLR-adaptor protein MAVS, indicative of MAVS activation.

2. ADAM9 is a proteolytic enzyme and cleavage of specific substrates may be involved to protect from EMCV infection. Although, unbiased substrate identification is rather complicated and probably beyond the scope of this work, the authors should at least use a catalytically inactive variant of ADAM9 to perform cell culture experiments to test if the catalytic activity is of relevance.

Studies with a catalytically inactive variant of ADAM9 are now shown in Figure 8 and demonstrate that ADAM9's ability to interact with MDA5 (**New Figures 8A and 8B**) and induce MDA5 oligomerization (**New Figure 8E**) does not require ADAM9's catalytic activity.

3. ADAM9 is known to cleave and solubilize the protease ADAM10, which has also been linked to certain viral infections. This issue should at least be discussed.

We have now discussed this in the Discussion section (lines 364_368). Moreover, in our binding assays of endogenous MDA5 and endogenous ADAM9 upon EMCV RNA stimulation, we also tested the interaction of ADAM10 as well as ADAM12 and ADAM17. This showed that whereas ADAM9 readily co-immunoprecipitated with MDA5, the other ADAM proteins did not (**New Figure 8D**).

4. Fig. 1B,C: What might be the reason that the viral titer decreases in ADAM9 knock-out mice 6 hours post-infection? We suggest that WT mice express ADAM9 on fibroblasts as well as unknown EMCV receptors on heart and brain cells. EMCV infects many different tissues including the pancreas, brain, and heart. The higher levels of EMCV in the serum of WT at 6hr post-infection may reflect faster kinetics of infection in WT animals as infection can occur via ADAM9 on parenchymal fibroblast and via the different tissue receptors for EMCV available to the virus in WT animals.

Reviewer #3 (Remarks to the Author):

It's a high quality and excellent manuscript. There are some minor questions:

1. Keywords section: as there was no experiment about mitochondrial antiviral-signaling protein (MAVS), so there is no need to include MAVS in keywords.

Thank you for pointing this out. We have now performed experiments testing the effect of ADAM9 on MAVS activation/oligomerization (**New Figure 8E**). We therefore left MAVS as a keyword.

2. Line 227, the format of 'β' should be checked and revised.

This has been corrected.

3. In the description of the results, it is easier for the reader to understand to write clearly figure A or B. e.g. 'ADAM9 mediates innate immune responses to picornavirus infection' section, only indicated as Figure 4 and Figure 5, each set of pictures has several small pictures, so it is easier for readers to understand when listed as Figure 4A, B, C or D clearly.

Thank you for raising this. This has now been corrected.

REVIEWERS' COMMENTS

Reviewer #2 (Remarks to the Author):

It was a pleasure to read the revised work. The authors address all of the points raised in detail. The additional data provided are compelling and the manuscript is suitable for publication.

Reviewer #3 (Remarks to the Author):

All my concerns have been addressed and I am satisfied with the revisions.